# InvariantStock: Learning Invariant Features for Mastering the Shifting Market

**Haiyao Cao**                                                         *haiyao.cao@adelaide.edu.au*
*Australia Institute for Machine Learning*
*The University of Adelaide*

**Jinan Zou**                                                           *jinan.zou@adelaide.edu.au*
*Australia Institute for Machine Learning*
*The University of Adelaide*

**Yuhang Liu**                                                         *yuhang.liu01@adelaide.edu.au*
*Australia Institute for Machine Learning*
*The University of Adelaide*

**Zhen Zhang**                                                        *zhen.zhang02@adelaide.edu.au*
*Australia Institute for Machine Learning*
*The University of Adelaide*

**Ehsan Abbasnejad**                                          *ehsan.abbasnejad@adelaide.edu.au*
*Australia Institute for Machine Learning*
*The University of Adelaide*

**Anton Van Den Hengel**                              *anton.vandenhengel@adelaide.edu.au*
*Australia Institute for Machine Learning*
*The University of Adelaide*

**Javen Qinfeng Shi**                                              *javen.shi@adelaide.edu.au*
*Australia Institute for Machine Learning*
*The University of Adelaide*

**Reviewed on OpenReview:** *https://openreview.net/forum?id=dtNEvUOZmA*

## Abstract

Accurately predicting stock returns is crucial for effective portfolio management. However, existing methods often overlook a fundamental issue in the market, namely, distribution shifts, making them less practical for predicting future markets or newly listed stocks. This study introduces a novel approach to address this challenge by focusing on the acquisition of invariant features across various environments, thereby enhancing robustness against distribution shifts. Specifically, we present InvariantStock, a designed learning framework comprising two key modules: an environment-aware prediction module and an environment-agnostic module. Through the designed learning of these two modules, the proposed method can learn invariant features across different environments in a straightforward manner, significantly improving its ability to handle distribution shifts in diverse market settings. Our results demonstrate that the proposed InvariantStock not only delivers robust and accurate predictions but also outperforms existing baseline methods in both prediction tasks and backtesting within the dynamically changing markets of China and the United States. Our code is available at https://github.com/Haiyao-Nero/InvariantStock

# 1 Introduction

In the realm of stock markets, portfolio optimization is a common technique aimed at maximizing profits while minimizing risks through the management of an asset basket. This approach, rooted in stock returns prediction, gained prominence with the introduction of the Capital Asset Pricing Model (CAPM) by Sharpe et al. [1964] and the Fama–French three-factor model Fama & French (1993). However, these traditional methods have often faced challenges in accurately predicting stock returns due to their inherent model simplicity. The landscape of research has witnessed a significant transformation with the advent of advanced deep-learning models. Many researchers now leverage deep learning techniques, including convolutional neural networks (CNN) LeCun et al. (1998); He et al. (2016), recurrent neural networks (RNN) Hochreiter & Schmidhuber (1997); Cho et al. (2014), and transformers Vaswani et al. (2017), for stock returns prediction. These methods are preferred for their exceptional fitting capabilities. Within modern deep learning frameworks, recent studies Zhang et al. (2017); Feng et al. (2018); Yoo et al. (2021); Duan et al. (2022); Gao et al. (2023); Zhao et al. (2023); Koa et al. (2023) have delved into various specific information sources within the stock market to enhance predictions.

Despite the advancements in deep learning techniques for stock return prediction, several limitations persist, particularly in the context of the scope and adaptability of these methods. A common issue is that many methodologies Zhang et al. (2017); Feng et al. (2018); Yoo et al. (2021); Gao et al. (2023) are constrained by either their inherent design or computational costs, leading to their application being limited to a small set of stocks, such as those in the DJIA, S&P 500, or CSI300 indices. This limitation inherently excludes a vast majority of market stocks, resulting in these models often capturing spurious correlations due to a lack of diversity in their training datasets. Consequently, these models tend to underperform or fail when faced with unseen or newly listed stocks. Moreover, the stock market is in a constant state of flux, influenced by a myriad of factors including political and economic environments. Regular occurrences like financial report announcements and public holidays, as well as unforeseen events such as wars or pandemics, significantly impact investor sentiment and market dynamics. Those factors result in an ever-shifting market distribution, a phenomenon also noted in the study of Zhao et al. (2023). Models that fail to account for these distribution shifts may prove impractical for future market applications Liu et al. (2022b). The recent work, FactorVAE Duan et al. (2022) has attempted to mitigate prediction noise by using dynamic factor models for cross-sectional return predictions. However, it does not take into account the market's distributional shifts. Similarly, Diffusion Variational Autoencoder(DVA) Koa et al. (2023) faces the same limitation. The DoubleAdaptr Zhao et al. (2023) attempts to incrementally train the model to adapt to the shifting market, it too encounters issues of catastrophic forgetting if the model is not retrained.

To effectively address the issue of distribution shifts in the stock market, we propose to identify features that remain consistent across varying distributions. Consequently, utilizing these features for prediction can offer robustness against distribution shifts across different environments. To achieve this goal, we frame it from an information theory perspective as follows: $H(Y|F) = H(Y|F, E)$, where $H$ denotes the entropy, $Y$ represents the target variable, $F$ signifies the invariant features across environments $E$. Essentially, this formulation implies that the objective is to learn features that can predict the target variable $Y$ with a high degree of accuracy while ensuring that these features remain unaffected by environmental factors within the stock market. Motivated by this, we proposed a designed learning framework, *InvariantStock*, including two main modules: an environment-agnostic prediction module and an environment-aware prediction module. In this framework, the first module is tailored to model $H(Y|F)$, while the latter is designed to model $H(Y|F, E)$. Through the designed learning process, the proposed framework can select features $F$ in such a way that $H(Y|F) = H(Y|F, E)$. Further, to enhance the implementation of the proposed framework, we have also devised an efficient selection module. This module plays a crucial role in ensuring that the selected features remain as consistent as possible. The contributions of this work can be summarized as follows:

- We address the challenges posed by distribution shifts in the stock market, by introducing an effective framework that centres on learning invariant features across different environments.

- We present InvariantStock, a designed learning framework that encompasses both an environment-agnostic prediction module and an environment-aware module, which facilitates the selection of

invariant features across diverse environments. We have designed an efficient selection module that seamlessly integrates into the proposed framework to further enhance the effectiveness of designed learning within the framework.

- Through experiments conducted using real-world market scenarios, we have demonstrated that the proposed InvariantStock framework surpasses other state-of-the-art prediction-based methods, in terms of a variety of evaluation metrics.

## 2  Related Work

**Regression-based stocks prediction**   In the field of stock prediction, regression-based methods represent a significant departure from traditional classification approaches. While classification typically involves sorting stocks into a limited number of categories, often two or three  Feng et al. (2018); Sawhney et al. (2020); Wang et al. (2021); Xiang et al. (2022), Regression-based methods  Duan et al. (2022); Zhao et al. (2023); Koa et al. (2023) aim to directly forecast the magnitude of price changes. This approach can be more informative and practical for investors, offering insights for decision-making processes like ranking predictions. Commonly, models based on CNN, RNN and transformer are employed for handling time series data in stock predictions  Wen et al. (2019); Feng et al. (2018); Yoo et al. (2021); Duan et al. (2022); Gao et al. (2023). Additionally, a growing body of research  Li et al. (2022b); Zhao et al. (2022); Zheng et al. (2023) has been exploring the use of graph-based models to capture the intricate relationships between stocks. However, prior work made predictions without considering the market shifting, which limits the performance and application in the real world.

**Feature selection**   In the domain of stock prediction, the necessity of feature selection is underscored by the need to minimize the impact of prevalent noise within financial data. Prior research in this field, including studies by  Tsai & Hsiao (2010); Shen & Shafiq (2020); Dona & Gallinari (2021); Haq et al. (2021); Li et al. (2022a) typically approaches feature selection through methodologies that prioritize features based on their assessed importance or their relational distance to the target variables. However, these conventional methods often overlook the dynamic nature of financial markets. In contrast, our features selection module is designed to identify a set of invariant features, those that retain their relevance and stability across varying market distributions.

**Invariant learning**   Invariant learning focuses on developing representations or models that remain consistent across various environments, utilizing contextual information to achieve this stability. The underlying rationale is that invariant features are closely linked to causality Liu et al. (2022a; 2024b;a). The methods for invariant seeking can be categorised into invariant risk minimization (IRM) Arjovsky et al. (2019); Chang et al. (2020); Koyama & Yamaguchi (2020) and domain-irrelevant representation learning  Ganin & Lempitsky (2015); Ganin et al. (2016). Our approach is inspired by the study of  Koyama & Yamaguchi (2020), which uses adversarial learning to do the IRM, Adapting this concept, we have developed a modified framework that incorporates a feature selection module.

## 3  Problem Formulation

This section outlines the setup and notation of the stock prediction problem, with a focus on stock return prediction combined with ranking. For the China stock market dataset, the prediction target is formulated as $y_t = \frac{p_{open}^{t+2} - p_{open}^{t+1}}{p_{open}^{t+1}}$, which also represents the ranking score, where $p_{open}^t$ denotes the opening price on day $t$. This formulation also serves as the ranking score. The rationale for using the opening price, rather than the closing price, is due to the regulatory limit on price ratio changes in the China stock market. Orders placed at market close may not be filled for stocks that have reached their price limit. However, orders placed at the opening are more likely to be filled, as few stocks reach their limit price at this time. In contrast, the prediction setup for the US stock market is more straightforward: $y_t = \frac{p_{close}^{t+1} - p_{close}^t}{p_{close}^t}$, This approach uses the closing price, with orders being placed as the market nears closing. The prediction task is represented as $\hat{y} = f(X, \Theta, \Phi)$ where $X = \{x_1, ... x_T\} \in \mathbb{R}^{T \times D}$ denotes the historical features of the stock. Here, $T$ is the

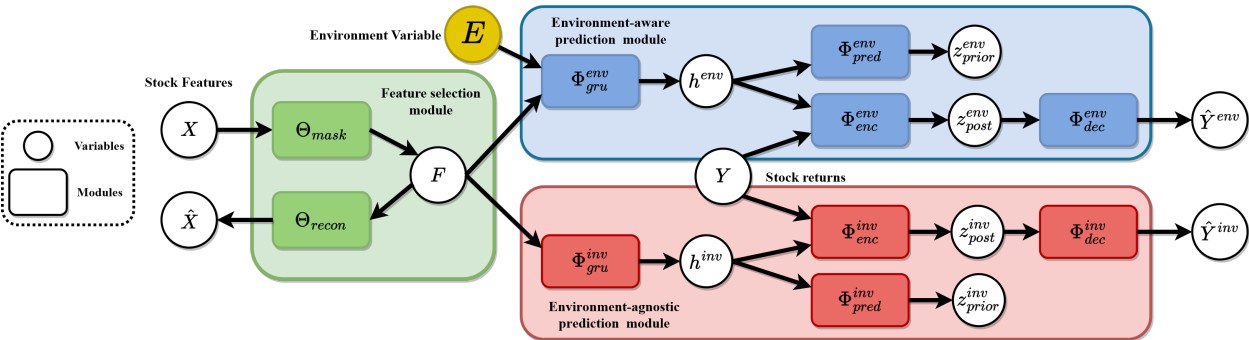

Figure 1: The structural design of InvariantStock is delineated, where the green region symbolizes the feature selection module, comprising a mask model and a reconstruction model. The red and blue regions depict the environment-agnostic prediction module and the environment-aware prediction module, respectively. Each prediction module is composed of a state extractor, an encoder, a decoder, and a predictor.

length of the look-back window, and $D$ represents the number of stock features at time $t$. $\Theta, \Phi$ symbolize the parameters of the feature selection module and the prediction module, respectively. Furthermore, given that environments continuously evolve over time, the primary focus of this work is to develop a learned model capable of delivering stable predictions despite these changes.

## 4    Methodology

The purpose of InvariantStock is to acquire the invariant features $F$ from the full features $X \in \mathbb{R}^{T \times D}$ by leveraging additional information provided by environmental variables, $E$. To construct $E$ we utilize a one-hot vector representation to encapsulate month and year data. This approach is grounded in the hypothesis that stock market distributions vary across different dates, thus implying that the market's characteristics and behaviour are not uniform. To learn the invariant features $F$, we introduce a feature selection module that utilizes a binary mask $M \in \{1,0\}^{T \times D}$, The purpose of this mask is to selectively filter the feature set such that the mutual information between the invariant features $F$ and $Y$ is maximized. This can be mathematically formulated as:

$$\max_{M \in \{1,0\}^{T \times D}} I(Y; F),$$
$$s.t.\ F = M \odot X, H(Y|F) = H(Y|F, E) \tag{1}$$

If it holds, $E$ does not provide any additional information beyond what is already captured by $F$ to predict $Y$. To achieve the objective in Equation 1, InvariantStock comprises three key components, which are depicted in Figure 1. Each with a specific function and objective: the Feature Selection Module $\Theta(X)$ which identifies crucial invariant features $F$; the Environment-Aware Prediction Module $\Phi^{env}(F, E)$, which incorporates environmental variables to model $H(Y|F, E)$ for context-sensitive predictions; and the Environment-Agnostic Prediction Module $\Phi^{inv}(F)$, focusing solely on stock features models $H(Y|F)$.

### 4.1    Prediction Modules

The prediction module in InvariantStock is designed with the primary objective of predicting stock returns based on $F$ and $E$. This module comes in two forms: the environment-aware prediction module and the environment-agnostic prediction module. The key distinction between these two lies in their access to the environmental variable $E$; the environment-aware module incorporates this variable, while the environment-agnostic module does not. InvariantStock leverages FactorVAE Duan et al. (2022) as the backbone of its prediction module, which comprises four components: a state extractor, an encoder, a decoder, and a predictor.
**State extractor:** GRU with attention is specifically chosen for its effectiveness in extracting meaningful

hidden states $h_t$ from historical data $x_t$ at time $t$. Formally,

$$h_t = \Phi_{GRU}(x_t, h_{t-1}) \tag{2}$$

**Encoder:** The encoder is responsible for deriving latent posteriors $z$ from the hidden state $h_t$ and the future ratio of price change $y$ by:

$$\begin{aligned}
[\mu_{post}, \sigma_{post}] &= \Phi_{enc}(y, h_t), \\
z_{post} &\sim \mathcal{N}(\mu_{post}, diag(\sigma_{post}^2))
\end{aligned} \tag{3}$$

where $\mu_{post}, \sigma_{post}$ is the distribution parameter of $z$.

**Decoder:** uses latent variables $z$ and hidden state $h$ calculate the stock return $\hat{y}$

$$\begin{aligned}
\alpha &\sim \Phi_{alpha}(h_t), \beta = \Phi_{beta}(h_t) \\
\hat{y} &= \alpha + \beta z
\end{aligned} \tag{4}$$

**Predictor:** To prevent future information leakage during the inference phase, the predictor is used to get the latent prior $z$ only from the hidden state $h_t$:

$$\begin{aligned}
[\mu_{prior}, \sigma_{prior}] &= \Phi_{pred}(h_t), \\
z_{prior} &\sim \mathcal{N}(\mu_{prior}, diag(\sigma_{prior}^2))
\end{aligned} \tag{5}$$

## 4.2 Feature Selection Module

The Feature Selection Module in InvariantStock is designed to select invariant features $F$ utilizing a generated binary mask $M$. This module's objective is not only to identify and isolate these invariant features but also to ensure that the original feature set $X$ can be accurately reconstructed from $F$. This dual functionality emphasizes retaining essential information while filtering out redundant features, thereby enhancing the model's predictive accuracy and reliability. Therefore, the feature selection module employs an AutoEnvoder(AE) structure where the $\Theta_{mask}$ and $M$ constitute the encoder and $\Theta_{recon}$ functions as the decoder a decoder. The equations governing this process are as follows:

$$M = \Theta_{mask}(X), F = M \odot X, \hat{X} = \Theta_{recon}(F) \tag{6}$$

In this setup, $\Theta_{mask}(X)$ generates the binary mask $M$ which is used to selectively enable or disable features in the historical sequence. Since the binary mask is not differentiable, the straight-through technique Bengio et al. (2013) is utilized to estimate the gradient. $\Theta_{recon}$ guarantees that all the invariant features are selected to avoid only a subset of invariant features being selected during the training.

## 4.3 Training Framework

InvariantStock achieves the goal in Equation 1 by employing multi-step training. Specifically, InvariantStock involves distinct objectives for each module: Environment-aware prediction module $\Phi^{env}$ and the environment-agnostic module $\Phi^{inv}$ are optimized by maxing the likelihood of predicting $Y$ conditioned on $F, E$ and $F$ respectively. Features Selection Module $\Theta$ targets identifying features that consistently influence stock returns, regardless of changing market environments. This is pursued by minimizing the discrepancy in predictions made by the environment-aware and environment-agnostic modules. Since there is no guarantee that all the invariant features could be learned only by minimizing the gap between two prediction modules, a reconstruction objective is introduced to retain all the invariant features instead of a subset during the learning process by recovering $X$ from $F$. These modules are trained sequentially, and the detailed training process for these objectives is systematically outlined in Algorithm 1. By assigning distinct, focused objectives to each component and adhering to this structured training protocol, InvariantStock is positioned to provide accurate and reliable predictions, adeptly handling the complexities of varying market scenarios.

**Prediction Objectives**

**Prediction Objectives** The objectives of the environment-aware module and environment-agonistic model are to maximize the conditional entropy and respectively by minimizing Equation 13 and Equation 12. The terms in these equations include minimizing the MSE loss of the targets and predictions while ensuring correct stock ranking (ordered by the targets) in the market. The predictions from these modules are defined as:

$$\hat{Y}^{env} = \Phi^{env}(X, E), \hat{Y}^{inv} = \Phi^{inv}(X) \tag{7}$$

Given the focus on selecting the most profitable stocks for portfolio selection, sample weights $W_t$ are introduced to emphasize stocks with significant price changes:

$$W_t = (\frac{rank(Y_t) - mean(rank(Y_t))}{std(rank(Y_t))})^2 \tag{8}$$

The prediction loss for both modules is calculated using Mean Square Error (MSE), factoring in these weights:

$$
\begin{aligned}
\mathcal{L}_{env}^{pred} &= \frac{1}{N} \sum_{i}^{N} (W_i Y_i - W_i \hat{Y}_i^{env})^2 \\
\mathcal{L}_{inv}^{pred} &= \frac{1}{N} \sum_{i}^{N} (W_i Y_i - W_i \hat{Y}_i^{inv})^2
\end{aligned}
\tag{9}
$$

Additionally, applying hinge loss for stock ranking ensures that the predicted rankings of stocks align closely with actual performance, which is crucial for selecting the right stocks in real trading. This method prioritizes accurate ranking, leading to better stock selection. The loss is defined as:

$$
\begin{aligned}
l_{env}^t &= \sum_{i=0}^{N} \sum_{j=0}^{N} \max(0, -(\hat{y}_{t,i}^{env} - \hat{y}_{t,j}^{env})(y_{t,i} - y_{t,j})) \\
l_{inv}^t &= \sum_{i=0}^{N} \sum_{j=0}^{N} \max(0, -(\hat{y}_{t,i}^{inv} - \hat{y}_{t,j}^{inv})(y_{t,i} - y_{t,j}))
\end{aligned}
\tag{10}
$$

The Kullback-Leibler Divergence (KLD) objective aims to minimize the difference between the posterior and prior distributions:

$$
\begin{aligned}
\mathcal{L}_{env}^{KL} &= -KL(P_{\Phi_{enc}^{env}}(z|h, y)||P_{\Phi_{pred}^{env}}(z|h)) \\
\mathcal{L}_{inv}^{KL} &= -KL(P_{\Phi_{enc}^{inv}}(z|h, y)||P_{\Phi_{pred}^{inv}}(z|h))
\end{aligned}
\tag{11}
$$

The total losses encompassing prediction accuracy, ranking effectiveness, and distribution alignment for both the environment-aware and environment-agnostic prediction modules are then calculated as:

$$\mathcal{L}_{env} = \mathcal{L}_{env}^{pred} + \alpha \sum_{t} W_t l_{env}^t + \beta \mathcal{L}_{env}^{KL} \tag{12}$$

$$\mathcal{L}_{inv} = \mathcal{L}_{inv}^{pred} + \alpha \sum_{t} W_t l_{inv}^t + \beta \mathcal{L}_{inv}^{KL} \tag{13}$$

**Feature Selection Objectives** The learning target of the feature selection module is to encourage the independence between $Y$ and $E$ given $F$ by minimizing the gap $\Phi_{env}$ and $\phi_{inv}$ so that $H(Y|F) = H(Y|F, E)$ is achieved. This objective is pursued through four strategies:
*Minimizing the Prediction Gap*: The module aims to reduce the discrepancy between the predictions made by the two modules. This is quantified using MSE:

$$\mathcal{L}_{\Theta}^{pred} = \frac{1}{N} \sum_{i}^{N} (W_i \hat{Y}_i^{inv} - W_i \hat{Y}_i^{env})^2 \tag{14}$$

---

**Algorithm 1** Training Process

---

**Input**: stock features $X$, Environment $E$ and stocks return $Y$
**Parameter**: Feature selection module $\Theta$, Environment-aware prediction module $\Phi_{env}$, Environment-agnostic prediction module $\Phi_{inv}$
**Output**:the prediction of stocks returns $\hat{Y}$

1: Let $e = 0$.
2: **while** $e <$ total epoches **do**
3:     $F = \Phi(X)$
    $\hat{Y}^{env}, \mu_{post}^{env}, \sigma_{post}^{env}, \mu_{prior}^{env}, \sigma_{prior} = \Phi_{env}(F, E, Y)$.
    $\hat{Y}^{inv}, \mu_{post}^{inv}, \sigma_{post}^{inv}, \mu_{prior}^{inv}, \sigma_{prior}^{inv} = \Phi_{env}(F, Y)$.
4:     **if** $e\%3 == 0$ **then**
5:         Calculate $\mathcal{L}_\Theta$ according to Equation 18.
        Update $\Theta$.
6:     **else if** $e\%3 == 1$ **then**
7:         Calculate $\mathcal{L}_{inv}$ according to Equation 13.
        Update $\Phi^{inv}$.
8:     **else if** $e\%3 == 2$ **then**
9:         Calculate $\mathcal{L}_{env}$ according to Equation 12.
        Update $\Phi^{env}$.
10:     **end if**
11: **end while**
12: **return** $\Theta$ , $\Phi^{inv}$

---

*Minimizing Ranking Objectives Difference*: This involves reducing the difference in the ranking objectives between the two prediction modules:

$$l_\Theta^t = \sum_{i=0}^{N} \sum_{j=0}^{N} \max(0, -(\hat{y}_{t,i}^{env} - \hat{y}_{t,j}^{env})(\hat{y}_{t,i}^{inv} - \hat{y}_{t,j}^{inv})) \tag{15}$$

*Minimizing KLD of Latent Variables*: This involves minimizing the difference in the distribution of latent variables generated by two prediction modules:

$$\mathcal{L}_\Theta^{KL} = KL(P_{\Phi_{pred}^{inv}}(z|h)) || P_{\Phi_{pred}^{env}}(z|h)) + KL(P_{\Phi_{enc}^{env}}(z|h,y) || P_{\Phi_{enc}^{inv}}(z|h,y)) \tag{16}$$

*Reconstruction Objective*: Ensuring that there's little information loss during selecting invariant features, we maximize the entropy $H(F|masked(F))$ through the following reconstruction objective:

$$\mathcal{L}_\Theta^{recon} = \frac{1}{NDT} \sum_{i}^{N} \sum_{j}^{DT} (x_{ij} - \hat{x}_{ij})^2 \tag{17}$$

Finally, the total learning objective of the feature selection module combines these elements:

$$\mathcal{L}_\Theta = \mathcal{L}_\Theta^{pred} + \alpha \sum_{t} W_t l_\Theta^t + \beta \mathcal{L}_\Theta^{KL} + \theta \mathcal{L}_\Theta^{recon} \tag{18}$$

This comprehensive objective reflects a multifaceted approach to feature selection, emphasizing the importance of accurate prediction, effective reconstruction, and alignment of latent variables. By addressing these aspects, InvariantStock's feature selection module aims to ensure robust and reliable stock return predictions.

## 4.4 Inference Progress

The proposed training framework, not required during the inference phase, ensures that the selected features remain invariant to environmental changes. During inference or in real-world applications, only the mask

module and the environment-agnostic prediction are retained. In contrast, the environment-aware module and the reconstruction model are utilized exclusively during the training process. As a result, the computational complexity and resource consumption during inference are significantly reduced compared to those during the training phase.

## 5 Experiments

To address the ensuing research inquiries:

- Does the InvariantStock model demonstrate efficacy on previously unobserved stocks within fluctuating market conditions? **(RQ1)**

- Is the applicability of the InvariantStock model consistent across diverse markets? **(RQ2)**

- What features predominantly contribute to the effectiveness of predictions in the shifting market? **(RQ3)**

### 5.1 Dataset

We conducted comprehensive assessments on both the China and the US stock markets, spanning more than 20 years. The details of these datasets are summarized in Table 1. we collect an extensive range of stock data, striving to accurately represent the real market conditions pertinent to portfolio selection. This extensive data collection was also aimed at minimizing potential biases in the dataset, thereby ensuring a more reliable and accurate assessment.

**China Stocks**  In assessing the ability to manage distribution shifts, The test set focuses on the period from early 2020 to 10/2022, a timeframe notably impacted by the COVID-19 outbreak. This global health crisis led to the widespread adoption of loose monetary policies, which, in turn, drove stock markets to new highs. However, by late 2021, the situation shifted due to mounting concerns about high inflation and expectations of rate hikes by the Federal Reserve. These factors contributed to a consistent downturn in the global financial market, reflected in the China stock market as well, as shown by the Shanghai Composite Index (Figure 5). Here, an initial uptrend before 02/2021 and a subsequent downtrend are evident, marking significant fluctuations in market conditions. The test set, comprising 1386 new stocks relative to the training set, presents an optimal scenario for evaluating the adaptability of our methods to these distribution shifts.

**US Stocks**  The approach to the US stock data was similar to that for the China stocks, with a comparable strategy for data segmentation. However, the US stock data is characterized by a more limited set of features, specifically six: open, high, low, close, volume, and the ratio of price change.

| | Set | #Date | #Stocks | #Features |
|---|---|---|---|---|
| **China** | Train | 05/1995-12/2016 | 2662 | **26** |
| | Validation | 01/2017-12/2019 | 3440 | |
| | Test | 01/2020-10/2022 | 4048 | |
| **US** | Train | 01/1990-12/2018 | 4120 | **6** |
| | Validation | 01/2019-12/2020 | 4831 | |
| | Test | 01/2021-01/2024 | 6314 | |

Table 1: The temporal range, the number of stock entities, and the number of distinct features employed for the training, validation, and testing datasets for China and US stock markets.

### 5.2 Training setup

The experimental setup for our study was conducted using a GTX3090 GPU. The look-back window length is set as 20. Adam with a learning rate of 0.0005 is used as the optimizer and it's scheduled by One cycle scheduler. both $\alpha$ and $\beta$ and $\theta$ are set to 1. There's no further tuning for these parameters.

### 5.3 Baselines

In our analysis, we benchmarked InvariantStock against both the market standard and the latest machine learning (ML)-based methodologies. The comparisons are as follows:

- **Market Benchmark:** For an overarching gauge of market performance, we utilize the CSI300 index for the China market and the DJIA for the US market.

- **Dataset Benchmark:** Acknowledging that our datasets do not encompass all stocks in the markets, particularly in terms of historical data, we introduce a 'dataset benchmark'. In this benchmark, each stock is allocated an equal percentage of the total asset value in the portfolio. It is crucial to note that this benchmark does not factor in commission fees.

- **FactorVAE:** As outlined in Duan et al. (2022), FactorVAE is a method predicated on prior-posterior learning and utilizes a Variational Autoencoder (VAE) framework as detailed by Kingma & Welling (2013). This VAE framework also serves as the backbone for the prediction module in InvariantStock.

- **DoubleAdapt:** Proposed by Zhao et al. (2023), DoubleAdapt employs both a data adapter and a model adapter to diminish the impact of domain shifts. Additionally, it leverages meta-learning to progressively capture market evolution.

- **Diffusion Variational Autoencoder (DVA):** Introduced by Koa et al. (2023), the DVA predicts multi-step returns using a hierarchical VAE combined with a diffusion probabilistic model. For evaluation and portfolio backtesting in our study, we consider only the initial element of the predicted return sequence for each stock.

### 5.4 Investiment Simulation

Based on the predictions generated, we employ the $TopK$ strategy for portfolio construction, wherein each stock within the portfolio is assigned equal weight. The $TopK$ strategy involves selecting the top $k$ stocks with the highest predicted returns for a long position, while simultaneously shorting the $k$ stocks with the lowest predicted returns. It's important to note that in the China stock market, due to specific policies, only long positions are executed. In China stock market, there are regulatory mechanisms such as 'limit-up' or 'limit-down' rules. When a stock hits its limit-up, it becomes hard for further buying orders to be filled. similarly, hitting the limit-down prevents selling further orders. Conversely, in the US market, both long and short positions are feasible. Additionally, a commission fee is unilaterally charged at a rate of 0.0015 in both markets.

### 5.5 Evaluation Metrics

In assessing the performance of various stock prediction methodologies, several performance metrics are employed: Information Coefficient (IC), IC Information Ratio (ICIR) as mentioned in Du et al. (2021), Rank Information Coefficient (RankIC), Rank Information Coefficient Information Ratio (RankICIR), which are defined as:

- **Information Coefficient (IC)**: describe the correlation between targets and predictions, and is expressed as:

$$IC = corr(Y_t, \hat{Y}_t) \tag{19}$$

- **IC Information Ratio (ICIR)**: measures the consistency and reliability of the prediction, and is expressed as:

$$ICIR = \frac{mean(IC)}{std(IC)} \tag{20}$$

- **Rank Information Coefficient (RankIC)**: measures the correlation between the ranking of predictions and ranking of targets, and is expressed as:

$$RankIC = corr(Rank(Y_t), Rank(\hat{Y}_t)) \tag{21}$$

where $rank(Y_t)$ is the target ranking of stocks returns and $rank(\hat{Y}_t)$ signifies the predicted ranking of stock returns.

- **Rank Information Coefficient Information Ratio (RankICIR)**: measures the consistency and reliability of the predicted ranking.

$$RankICIR = \frac{mean(RankIC)}{std(RankIC)} \tag{22}$$

The backtesting metrics include annualized return rate (ARR), maximum drawdown (MDD), and Sharpe Ratio (SR). Higher ARR, higher SR, and lower MDD It is crucial to note that performance metrics are specifically applicable to prediction-based methods, including FactorVAE, DoubleAdapt, DVA, and InvariantStock, while Metrics such as ARR, MDD, and SR are primarily utilized for evaluating the backtesting effectiveness of the compared methods.

### 5.6 Performance on China Stock market (RQ1)

The performance results of the compared methods are shown in Table 2 and Figure 2.

| Method | IC | ICIR | RankIC | RankICIR | ARR | MDD | SR |
|---|---|---|---|---|---|---|---|
| **CSI300** | - | - | - | - | -0.035 | -0.3902 | -0.2988 |
| **Dataset benchmark** | - | - | - | - | 0.1257 | -0.2828 | 0.4625 |
| **FactorVAE** | 0.0331 | 0.5769 | 0.0447 | 0.6627 | 0.0057 | -0.3048 | -0.0825 |
| **DoubleAdapt** | 0.0376 | 0.4911 | 0.0707 | 0.9482 | 0.1701 | -0.2668 | 0.6037 |
| **DVA** | 0.0002 | 0.0119 | 0.0005 | 0.0336 | -0.1919 | -0.5352 | -0.9803 |
| **InvariantStock** | **0.0576** | **0.6896** | **0.1005** | **1.0900** | **0.8315** | **-0.1878** | **3.7198** |

Table 2: The performance of predictions and backtesting outcomes for the methodologies compared within the Chinese stock market.

InvariantStock distinctly outshines all other methods in the comparative analysis, achieving the highest performance across both performance and backtesting metrics in the China stock market. The superiority of InvariantStock over FactorVAE is evident through the integration of a Feature Selection Module and diverse learning objectives, effectively serving as an ablation study for the feature selection process. Therefore this comparison not only underlines the model performance but also highlights the critical role of targeted feature selection in improving functionality. Meanwhile, the return trend of DoubleAdapt aligns more closely with the dataset benchmark. Notably, the DVA method fails to deliver effective results in this context.

These findings underscore the capability of InvariantStock to adeptly navigate the distribution shifts characteristic of China stock market, a feat that other baseline methods struggle to accomplish. As evidenced by the backtesting performance after 02/2021, the cumulative return stays increasing (Figure 2) even with a declining trend of SSEC (Figure 5). While most methods face challenges in achieving commendable performance amidst market shifts and with unseen stocks, InvariantStock demonstrates robust performance in both uptrend and downtrend market scenarios by selecting the features that encourage the independence between the environment and stock return.

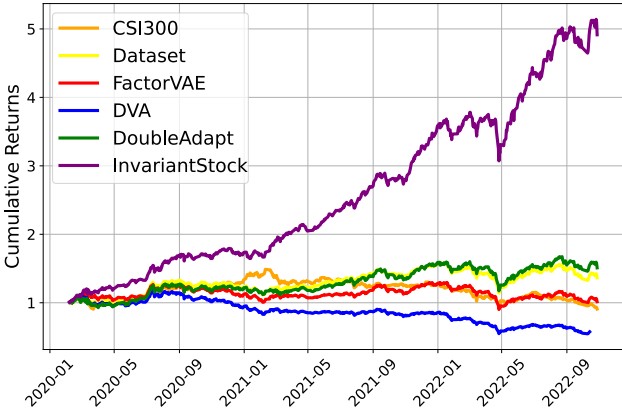

Figure 2: Comparative analysis of cumulative returns by different methods in the China stock market highlighting the superior performance of InvariantStock amidst market shifts.

## 5.7 Performance on US stock market (RQ2)

The performance outcomes of the methods under comparison are presented in Table 3 and Figure 3. InvariantStock, when evaluated on performance metrics, does not yield results comparable to those observed on the China dataset. This disparity may stem from the limited variety of stock features in the US dataset, which primarily includes only six price-related features. Nevertheless, InvariantStock still demonstrates superior practical performance in terms of RankIC and RankICIR compared to other baseline methods. Moreover, the backtesting results on ARR and SR for InvariantStock remain more commendable than the others. This suggests that InvariantStock is capable of picking profitable stocks even though the overall performance is not good. While DoubleAdapt also shows commendable results, but not as good as InvariantStock's. The ARR of FactorVAE is slightly better than that of the DJIA. DVA method does not exhibit effective performance in the US market either. It is noteworthy that the MDD of all the proposed methods is inferior when compared to the DJIA, which means all the methods cannot handle the risk well in the US market. This discrepancy may be caused by the lack of features in the US market, we make further analysis in Section 5.8.

| Method | IC | ICIR | RankIC | RankICIR | ARR | MDD | SR |
|---|---|---|---|---|---|---|---|
| **DJIA** | - | - | - | - | 0.0810 | **-0.2194** | 0.3657 |
| **Dataset** | - | - | - | - | -0.0550 | -0.3773 | -0.3576 |
| **FactorVAE** | 0.0085 | 0.0757 | -0.0031 | -0.0208 | 0.0953 | -0.4461 | 0.2343 |
| **DoubleAdapt** | **0.0114** | **0.1270** | -0.0127 | -0.1169 | 0.3192 | -0.3422 | 1.2788 |
| **DVA** | -0.0013 | -0.0866 | -0.0009 | -0.0636 | -0.3427 | -0.7109 | -4.7056 |
| **InvariantStock** | 0.0090 | 0.0840 | **0.0021** | **0.01605** | **0.5810** | -0.3107 | **1.9119** |

Table 3: The performance of predictions and the backtesting outcomes for the methodologies compared within the US stock market.

## 5.8 Invariant Features Selection (RQ3)

We computed the mean values of the masks applied to the China test set, as depicted in Figure 4. Within this representation, certain features exhibit a light hue, indicating their prominence while the dark hue is opposite. The less important features include the open price [0], high price [1], close price [3], last close price [4], percentage of change [6], total share [19], and next open price [24]. A comprehensive list of all features along with their corresponding indexes is detailed in Table 5. The analysis reveals that most of the lighter features in the dataset are fundamental features. This observation leads to the conclusion that

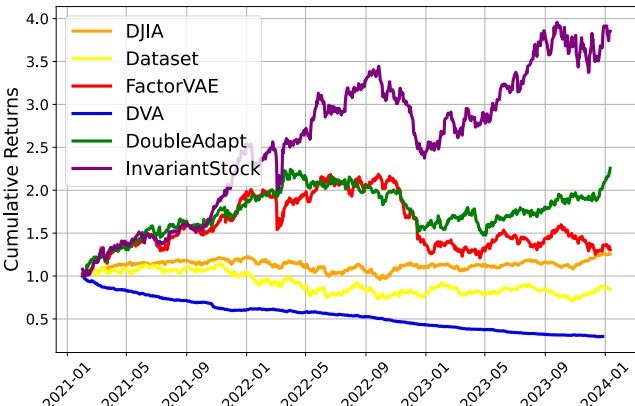

Figure 3: The cumulative returns from various methods in the US Stock Market indicate that with a strategically chosen number of stocks in the portfolio, approaches like InvariantStock, DoubleAdapt, and FactorVAE are capable of yielding improved results. However, these methods consistently encounter a higher maximum drawdown, suggesting an increased risk factor in their performance.

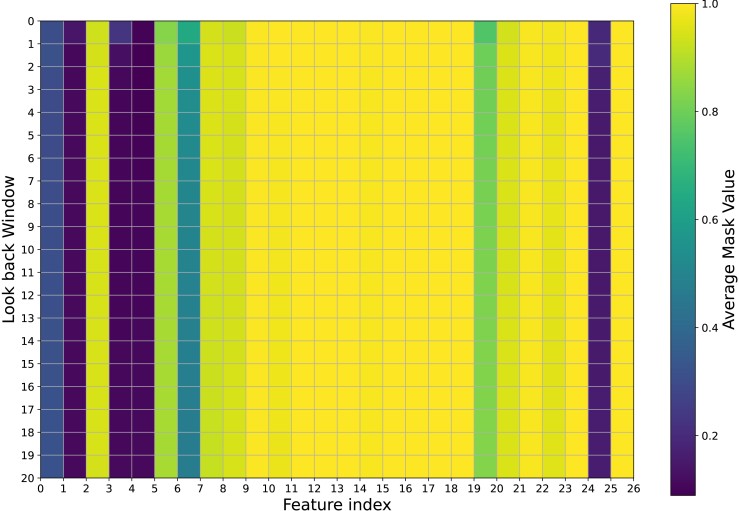

Figure 4: Visualization of the mean value of the mask across all test data, indicating feature significance. Lighter colours represent features of higher prominence. The features are specified in Table 5. The selection model demonstrates a preference for most fundamental features, while price-related features are generally disfavored.

fundamental features play a crucial role in stabilizing predictions across different market distributions, while price features tend to introduce volatility in predictions. The feature selection module of InvariantStock effectively identifies and utilizes these invariant features, contributing to stable stock return predictions. Since there are only price features in the US dataset, this finding explains why InvariantStock does not yield comparable results as in the China dataset.

Additionally, The lookback window is set to 20 days, An important observation from this setting is the relatively consistent colouration across the time dimension in the feature representation. This consistency suggests that each feature contributes equally to the prediction over this 20-day period, indicating a stable influence of each feature within the look-back window. A particularly intriguing finding is the notable significance of the low price in contributing to stable predictions, which is worth further investigation.

Furthermore, we examined the most profitable transactions, as illustrated in Table 5. The analysis shows that the mask varies across different dates, with more similarity observed in closer dates. Notably, most profitable transactions occur in the Second-board Market, identifiable by stock codes beginning with '300'. This trend is attributed to the more lenient price limitations in the Second-board Market compared to the Main-board Market.

### 5.9 Ablation Study

**Weighted Mask Vs Binary Mask**   We test weighted and binary mask methods for the feature selection on the China market, and the results are shown in Table 4. We can see that their performance (IC, ICIR, RankIC and RankICIR) on the test set are close. However, the backtest results (ARR, MDD, and SR) are different, the result by binary mask is better than that by weighted mask. The reason could be the use of the TopK strategy for the portfolio selection. Although the performance of the two methods is close, the performance of the selected stocks is different, which leads to the difference in the backtesting.

| Method | IC | ICIR | RankIC | RankICIR | ARR | MDD | SR |
|---|---|---|---|---|---|---|---|
| **Binary mask** | **0.0576** | **0.6896** | 0.1005 | **1.0900** | **0.8315** | **-0.1878** | **3.7198** |
| **Weighted mask** | 0.0552 | 0.6492 | **0.1007** | 1.0522 | 0.1953 | -0.4752 | 0.4858 |

Table 4: The performance of predictions and the backtesting outcomes on China market for the different feature selection techniques, including weighted mask and binary mask.

**Portfolio Size**   In both China market and US market, we examine how the Sharpe Ratio of compared methods fluctuates with the number of stocks included in the portfolio, as depicted in Figure 8. It is observed that Invariant consistently surpasses other methods, regardless of the portfolio size. This consistent outperformance further substantiates the robustness of the InvariantStock approach. Moreover, the portfolio with 100 stocks could achieve the best performance, which means it a relatively more stable than the best portfolio with fewer stocks in the US market when facing risks. With the increase in the number of stocks included in the portfolio, the performance of all the methods tested tends to converge towards the dataset benchmark. That's also why we introduce the dataset benchmark into the comparisons.

## 6   Limitations and future work

In this study, we focus on learning invariant features across different environments, with the model showing a preference for fundamental features over price-related ones. However, the exclusion of certain price features may result in a loss of predictive capability. Consequently, if some confounding variables between the target and features are unobservable, the model may tend to discard these features to maintain stable predictions across varying environments, thereby limiting the model's performance. Therefore, exploring and identifying the latent variables from the observational ones could bridge this gap, which is worth studying in the future. Another potential future direction involves the selection of environmental variables. In this work, we utilized the date index as the environmental variable. However, market conditions, asset types, and geographical regions could also be considered as alternative environmental variables for stock prediction and are worth exploring.

## 7   Conclusion

In this study, we elucidate the phenomenon of market distribution shifts resulting from alterations in policy and economic landscapes. We introduce a learning framework named InvariantStock, designed to master these distribution shifts by learning invariant features. Rigorous experimentation on previously unseen stocks substantiates the resilience and efficacy of InvariantStock. Additionally, backtesting both the Chinese and United States stock markets reveals that InvariantStock attains unparalleled performance, affirming its consistent reliability. Intriguingly, our findings indicate that robust stock prediction hinges more significantly

on fundamental features as opposed to those related to stock prices. However, it is noteworthy that the portfolio in this study was arranged in a basic manner, suggesting that future research could potentially enhance this aspect for further refinement.

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

## A   The Trend of SSEC

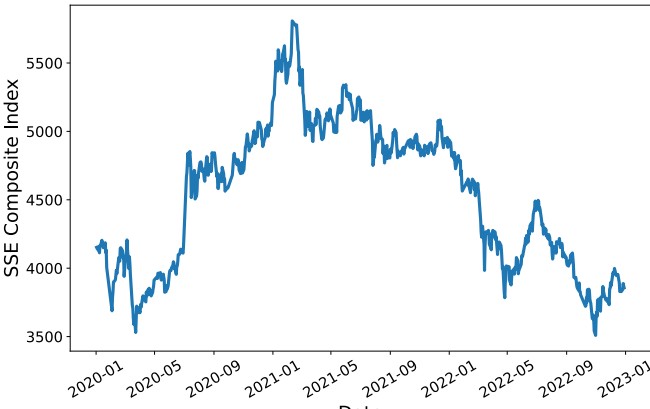

Figure 5: The historical trend of SSEC depicting the distribution shifting in China stock market.

## B   Ablation Study

Figure 6 and Figure 7 are the sharp ratios from the backtesting with different numbers of stocks on the China stock market and the US stock market.

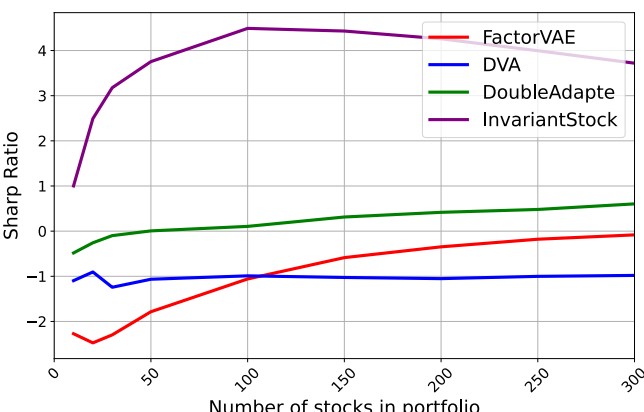

Figure 6: The SR results from backtesting various numbers of stocks in the China stock market reveal a notable trend. Specifically, when the portfolio comprises 100 stocks, the SR achieved by InvariantStock is the highest among the methods tested. In comparison, the SRs from other baseline methods lag significantly behind InvariantStock's, underscoring its superior performance in this context.

## C   Features Analysis

Table 5 is the features in the China dataset and the corresponding index. Figure 8 is the feature mask of best cases during the backtesting.

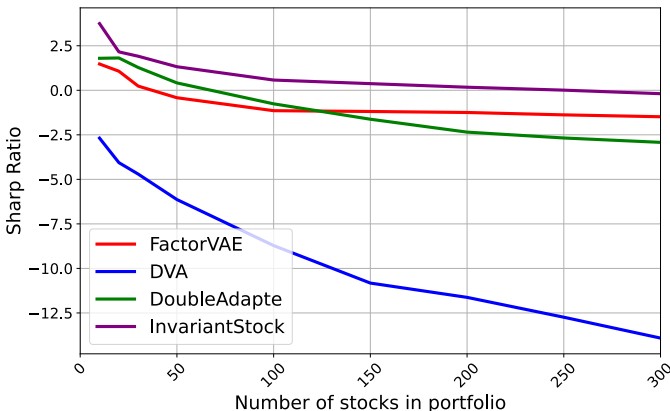

Figure 7: The Sharpe Ratio (SR) results from backtesting with varying numbers of stocks in the US stock market show that InvariantStock continues to outperform other methods, albeit the margin of superiority is not as pronounced as observed in the China market. This suggests that while InvariantStock maintains a lead in performance, the competitive gap between it and other methods is narrower in the US market context.

| Index Feature | 0 Open | 1 High | 2 Low | 3 Close | 4 Previous close | 5 Price change | 6 Percentage of change |
|---|---|---|---|---|---|---|---|
| Index Feature | 7 Volume | 8 Amount | 9 Turnover rate | 10 Turnover rate (CMC) | 11 Volume ratio | 12 PE | 13 PE (TTM) |
| Index Feature | 14 PB | 15 PS | 16 PS (TTM) | 17 Dividend yield | 18 Dividend yield (TTM) | 19 Total share | 20 Circulated share |
| Index Feature | 21 Free circulated share | 22 Market Capitalization | 23 Total Capitalization | 24 Next open | 25 Overnight price change | | |

Table 5: Features and its corresponding index in China dataset, Circulated Market Capitalization(CMC) Trailing Twelve Months(TTM) Price-to-sales Ratio(PS) Price/book value ratio(PB) Price-earning ratio(PE). Among those, [0-11,24,25] are the price features or technique indicators, which are decided by the trading between investors, while [12-23] are the fundamental features that reflect the decision of the company or the financial situation of the company.

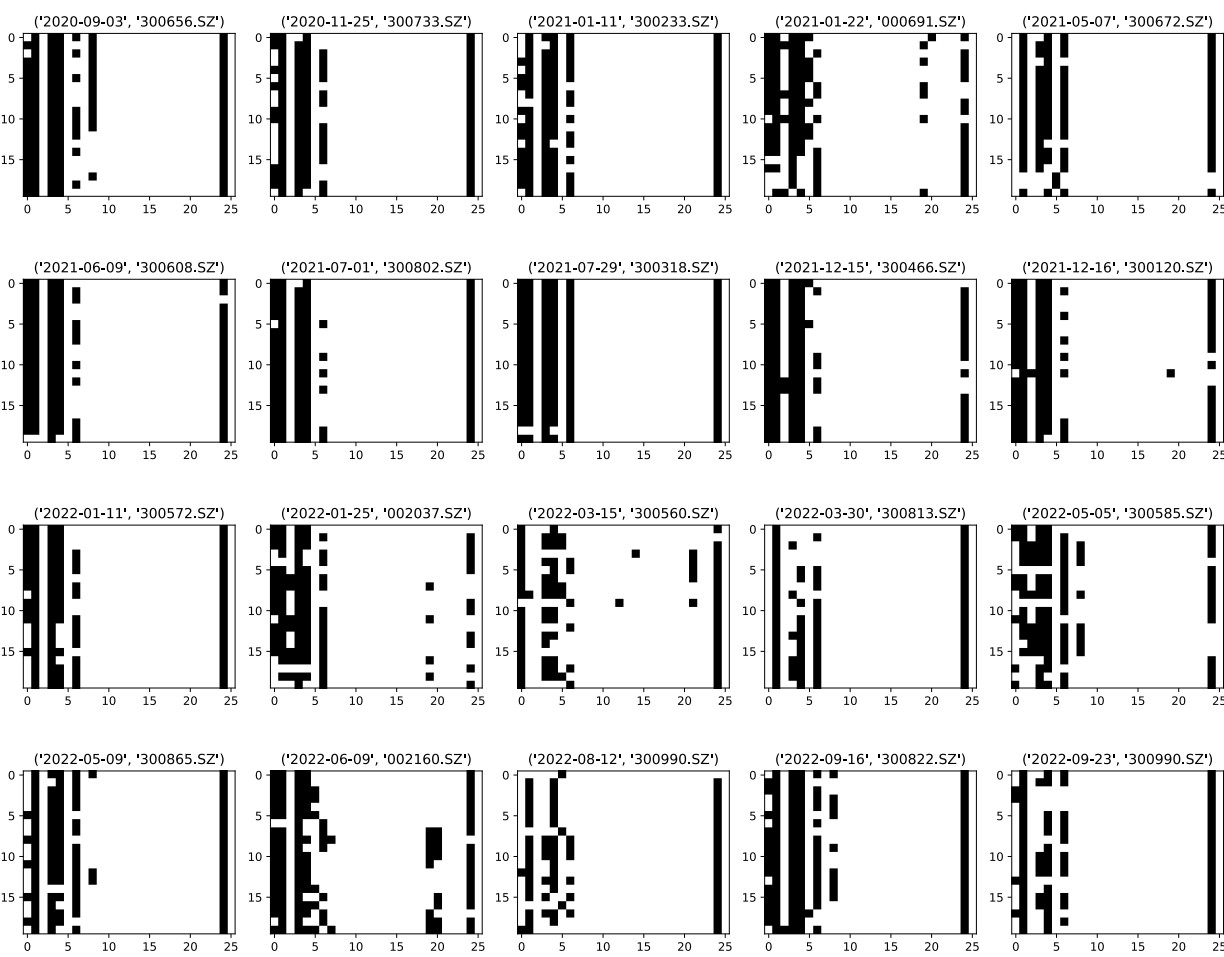

Figure 8: The most profitable transactions made by InvariantStock in China stock market. the position filled in black means the masked features.

