# OpenReview forum: "InvariantStock: Learning Invariant Features for Mastering the Shifting Market"
_TMLR — Accepted by TMLR_

### Review · Reviewer_ecVD · 2024-05-31

**Summary Of Contributions:**

This paper addresses the significant and impactful challenge of predicting stock returns amidst distribution shifts, presenting a well-focused research scope. The proposed approach effectively learns invariant features that enhance robustness and prediction accuracy in the stock market. The innovative framework design comprises an environment-agnostic prediction module and an environment-aware prediction module, facilitating the learning of invariant features across different environments. The effectiveness of the framework is empirically validated through experiments in real-world market scenarios, demonstrating that InvariantStock outperforms state-of-the-art methods in various evaluation metrics.

**Audience:**

Yes

**Broader Impact Concerns:**

The technique may be limited to scenarios where a large number of stock data features are available and may not be applicable to scenarios with only a limited number of features.

**Claims And Evidence:**

Yes

**Requested Changes:**

- Add Needed Technical Justifications: Include thorough technical justifications for design choices such as the use of binary masks instead of weighted ones, and the selection of hinge loss. These justifications should be supported in the ablation study section to demonstrate their impact on the model’s performance.
- For experiments, there are a couple of changes needed to be made.
(a) Data Processing and Parameter Choices: Provide a detailed discussion on how the training data is processed and the rationale behind parameter choices.
(b) Expanded Testing Cases: Include more diverse cases for testing to demonstrate the model’s robustness across different scenarios.
(c) Impact of Original Number of Features: Test and analyze how the original number of features affects the results, providing insights into the model’s sensitivity to feature selection.
(d) Explanation of Experimental Results: Improve the explanation of experimental results by discussing what the metric improvements mean in practical terms, rather than merely presenting a table of results.
- Reflect Novelty: Extend the discussion of related work to better reflect the novelty of the current research. Clearly articulate the fundamental research contributions of this work compared with previous studies. Expand the discussion on feature selection and invariant learning. Clarify the types of features referred to in this work, specifying whether statistical features like moving averages or other economic indicators such as GDP and EPS are considered.
- Improve Presentation
(a) Problem Formulation: Enhance the presentation of the paper by clearly formulating the problem in Section 3. Move beyond basic background definitions to a detailed discussion of the specific problem being addressed.
(b) Equation Explanations: Provide a thorough explanation of Equation 1, including the concept of mutual information between the invariant features F and the target variable Y. Clearly define and explain the H function and its significance in the context of the paper.
(c) Explain how E is constructed.

**Strengths And Weaknesses:**

Strengths:
- Important and Impactful Research Problem: The paper addresses a significant and impactful research problem with a well-focused research scope, specifically the challenge of predicting stock returns amidst distribution shifts.
- Invariant Feature Learning: The approach effectively learns invariant features that can handle distribution shifts in the stock market, enhancing robustness and prediction accuracy.
- Framework Design: The framework design comprises an environment-agnostic prediction module and an environment-aware prediction module. This dual-module design facilitates the learning of invariant features across different environments.
- Empirical Validation and Strong Performance: The framework’s effectiveness is validated through experiments in real-world market scenarios, demonstrating the strong performance of InvariantStock compared to state-of-the-art methods.

Weaknesses:
- Insufficient Technical Justifications: The paper lacks comprehensive technical justifications for all the design choices made in the proposed framework.
- Weak Evaluation: The evaluation of the proposed method is not robust enough to convincingly demonstrate its effectiveness.
- Insufficient Discussion of Related Work: There is a lack of thorough discussion of related work, which would provide better context and highlight the contributions of this study.
- Hard-to-Follow Presentation: The presentation workflow is difficult to follow in some sections, which affects the clarity and readability of the paper.

---

> ### Author Response · Authors · 2024-07-19
>
> ### Technical Justifications
> Thank you for your suggestions. We conduct experiments using a weighted mask, and the performance is shown in the table below. Although the average performance on the test dataset is similar to that of the binary mask, its backtest results are inferior compared to the binary mask. The reason is that we use the TopK strategy for portfolio construction, the performance of the selected stock by binary mask is better than that by weighted mask, though the average performance is close.  therefore, the stocks selected using binary masks are significantly more profitable than those selected using weighted masks.
> |Mask Type|IC|ICIR | RankIC |RankICIR|ARR |MDD |SR |
> | :-----:| ----: | :----: |:----: |:----: |:----: |:----: |:----: |
> | __Binary mask__ |__0.0576__|__0.6896__| 0.1005 |__1.0900__|__0.8315__|__-0.1878__|__3.7198__ |
> |__Weighted mask__ | 0.0552 |0.6492 |__0.1007__ |1.0522|0.1953 |-0.4752 | 0.4858 |
>
> As for the reason for choosing hinge loss, it is because our model is ultimately intended for portfolio management. Given that the goal of portfolio management is to select stocks based on predictions using the TopK strategy, we introduce hinge loss to encourage the predictions to have an accurate order.
>
> ### Experiments
> Thanks for your suggestion we will make the changes according to the clarification below.
> + For the dataset, the features we use are presented in Table 4 (supplementary). We do not apply any additional data preprocessing techniques except for normalization. For the parameters, we use the same hyperparameters as those in the previous work FactorVAE [1], based on an unofficial implementation available at \url{https://github.com/x7jeon8gi/FactorVAE} (not developed by us). Therefore, our framework has demonstrated a strong performance even without extensive hyperparameter tuning.
> + The test period begins in early 2020, coinciding with the outbreak of COVID-19. During this period, there are two main phases: an obvious uptrend from January 2020 to January 2021 and an obvious downtrend from February 2021 to October 2022. In both phases, our model consistently achieves the best cumulative returns. These distribution shifts are illustrated in Section 5.1 and Figure 5.
> + Regarding the impact of features, we present their significance in Section 5.8 and Figure 4. Our model tends to select fundamental features (e.g., PE, PB, and PS) over price features (e.g., Open, High, Low, and Close). Additionally, because the US dataset we collected does not contain fundamental features, the model does not achieve results comparable to those obtained with the China dataset.
> + We follow the evaluation metrics used in previous studies such as FactorVAE and DeepAdapt. We will also provide the evaluation metrics definition and details in the published version. Annualized Return Rate (ARR), Sharpe Ratio (SR) and Maximum Drawdown (MDD) are the common metrics for evaluation in finance trading. the high return rate indicates that the model performs well, while the appearance of drawdowns represents mistakes made by the model. From the test results, we can see that InvariantStock consistently achieves the highest ARR, the highest SR, and the lowest MDD, which highlights the benefits of our framework.
>
> ### Reflect Novelty
> We have outlined the novelty and contributions of our method in the introduction section and we will discuss them in detail in the related work section compared to previous studies. The features can be classified into two categories: technical indicators and fundamental indicators. We will clarify this in the published version. While macro indicators (e.g., GDP or CPI) are not included in the features, EPS (Earnings Per Share), which can be calculated using stock price, PE ratio, and total shares, is implied by the features. However, macro indicators have the potential to serve as environmental variables, and we will explore this direction in future work.
>
> ### Presentation Improvements
> Thank you for your suggestions. The content you mentioned is addressed in this paper, and we will reorganize some content to enhance clarity.
> + The task of prediction is to forecast future price changes, but the stock market distribution varies across different periods specified by year and month (as discussed in Section 4). We will emphasize how market shifts impact prediction tasks in Section 3.
> + The objectives of the environment-awareness module and environment-agonistic model are to maximize the conditional entropy $H(Y|X,E)$ and $H(Y|X)$ respectively by minimizing Equation (12) and Equation (13). The terms in these equations include minimizing the MSE loss of the targets and predictions while ensuring correct stock ranking (ordered by the targets) in the market.
> + The construction of $E$, which changes across different periods and is specified by year and month, is introduced in section 4.

---

### Review · Reviewer_dtfG · 2024-06-21

**Summary Of Contributions:**

The paper introduces a new approach to learn invariant features for dealing with stock market shifting. They use an information theoretical criterion in helping select the feature sets which are invariant to environmental factors. The experimental results show the effectiveness of the proposed approach.

**Audience:**

Yes

**Broader Impact Concerns:**

The approach has the potential to be adopted and further studied by industry and academics related to finance and AI.

**Claims And Evidence:**

Yes

**Requested Changes:**

1. I would like to see some in-depth discussions on discovering the invariant features, what technical challenges are there and how you solve it?
    What makes this problem unique and challenge if any?

2. Can this idea be generalized to other settings, the invariant features should have applications beyond stock market. Can you find any?

**Strengths And Weaknesses:**

Strengths:
1. The problem is both practical and interesting.
2. The approach is simple and easy to implement
3. The experimental results demonstrate the effectiveness of the proposed approach

Weaknesses:
1. The overall technical strength is low; most of the technical details are simply adopted from other papers or rather standard techniques
    For instance, most of the contents in page 4, 5, 6.
2. The technical highlight is not clear, and it doesn't delve into too much on the invariant feature discovery and what it takes to get them.
3. For this type of research, I am a bit worried about its practical and theoretical contribution: for the practical impacts, the industry likely adopt much more sophisticated approaches; for the theoretical side, it lacks of deep technical merits.

---

> ### Author Response · Authors · 2024-07-19
>
> ### The challenges and our proposed solution
>
> In the fintech domain, previous research often fails to account for market distribution shifts when predicting stock price changes, potentially leading to significant losses when such shifts occur. In contrast, we introduce a feature selection module designed to exclude variables that fluctuate with the market but exhibit a spurious relationship with stock price changes, thereby achieving independence between the targets and environmental variables conditioned on the selected features. Moreover, the feature selection module is optimized by minimizing the predictions of both an environment-awareness module and an environment-agnostic module.
>
> ### Conceptual framework and applicability of our method
> The key idea of this paper is to select features on which the target variable and the environment variable are conditionally independent. However, if we set different targets or environments, the selected features will vary to maintain independence between the target and environment. Though the applications in other domains are beyond the scope of our work, this framework can also be applied to other scenarios where we aim to ensure that the model's predictions conditionally remain invariant across different environmental distribution shifts.

---

### Review · Reviewer_vRBt · 2024-06-24

**Summary Of Contributions:**

This work introduces InvariantStock, a learning-based framework that addresses distribution shifts in stock markets by focusing on invariant feature learning across various environments.

A feature selection module is integrated, designed to identify features that remain consistent across different market distributions, thereby enhancing predictive stability and accuracy.

One interesting idea from this work is to use maximizing mutual information between invariant features and the target variable while ensuring robustness against environmental variability.

**Audience:**

Yes

**Broader Impact Concerns:**

No concerns.

**Claims And Evidence:**

Yes

**Requested Changes:**

No requested changes, some optional adjustments.

- Conduct additional experiments across different market conditions, asset types, and geographical regions to further validate the framework's robustness and generalizability.


- To improve performance in markets with limited feature diversity, consider integrating additional sources of data (e.g., macroeconomic indicators, sentiment analysis from news articles, etc.) to enrich the feature set.

**Strengths And Weaknesses:**

## Pros

1. The introduction of invariant feature learning to address distribution shifts is a significant advancement in stock return prediction methodologies.

2. The combination of environment-aware and agnostic modules provides a robust mechanism for feature selection and prediction.

3. Empirical performance is strong to show evidence of the framework's effectiveness.

4. The feature selection module ensures that only the most relevant and consistent features are used.

## Cons

1. The framework's performance in the US market was notably less impressive than in the China market, indicating potential issues with generalizability across different datasets.

2. The higher MMD compared to market benchmarks suggests the framework may not effectively manage risk in volatile / OOD conditions.

3. The multi-step training process and complex modules might lead to significant computational overhead, limiting real-time applicability.

4. Heavy reliance on historical data could be a limitation in highly volatile or unprecedented market conditions, where past data may not be indicative of future trends.

5. The binary mask and straight-through technique used in feature selection is possible with iterative alignments.

---

> ### Author Response · Authors · 2024-07-19
>
> ### Optional adjustments
> Thank you for your suggestion. The adjustments you mentioned above are also two future directions we intend to explore.
> We concur that market conditions, asset types, and geographical regions could serve as environmental variables. In future work, we will investigate how these different environmental variables influence the stock market. Additional features, such as macroeconomic indicators, news, financial announcements, and different resolution data (e.g., minute-level data or tick data), have significant potential to elucidate stock price fluctuations.
>
> ###  Performance disparity between the US and China markets
> Additionally, the performance disparity between the US and China markets can be attributed to the differing number of features in their respective datasets. The China datasets contain 26 features, whereas the US datasets contain only 6 features. Consequently, the limited number of features in the US datasets does not provide sufficient explanatory power for stock price changes, resulting in the MMD of our method being larger than the benchmark in the US market (Section 5.8).
>
> ### Computation problem.
> For real-time application during the inference phase, only the mask module and the environment-agnostic module are required, while the environment-aware module and the reconstruction model are required only during the training process. We will make it clear in the published version.
>
> ### Alternatives for straight-through technique
> We also acknowledge that the binary mask and straight-through technique have alternatives, such as the weighted mask. We conducted the experiments using weighted mask for feature selection techniques and the compared result is shown in the table below, which will be added to the published version. The table below shows the comparison results, where we can see that their performance (IC, ICIR, RankIC and RankICIR) on the test set are close. However, the backtest results (ARR, MDD, and SR) are different, the result by binary mask is better than that by weighted mask. The reason could be the use of the TopK strategy for the portfolio selection. Although the performance of the two methods is close, the performance of the selected stocks is different, which leads to the difference in the backtesting.
>
> |Mask Type|IC|ICIR | RankIC |RankICIR|ARR |MDD |SR |
> | :-----:| ----: | :----: |:----: |:----: |:----: |:----: |:----: |
> | __Binary mask__ |__0.0576__|__0.6896__| 0.1005 |__1.0900__|__0.8315__|__-0.1878__|__3.7198__ |
> |__Weighted mask__ | 0.0552 |0.6492 |__0.1007__ |1.0522|0.1953 |-0.4752 | 0.4858 |

---

> > ### Comment · Reviewer_vRBt · 2024-08-09
> >
> > Thanks the authors for the clarification. I don't have further questions straight-through ablations.

---

### Comment · Reviewer_vRBt · 2024-06-19
**Official Comment**

- Summary

The paper presents a learning based framework, which is designed to address the issue of distribution shifts in stock markets by learning invariant features across different environments. An environment-aware prediction module and an environment-agnostic prediction module have been introduced.

The framework is theoretically motivated by information theory, specifically aiming to maximize the mutual information between invariant features and the target variable while ensuring these features are unaffected by environmental variables. The study highlights the practical applicability of InvariantStock in dynamic market conditions, showing its robustness in predicting stock returns and outperforming estimation based and recent ML-based methods.

- Technical Weaknesses

The backtesting results showed that all methods, including InvariantStock, had a higher maximum drawdown compared to the market benchmarks, indicating that the proposed framework might not effectively handle risk in volatile market conditions.

The feature selection module uses a binary mask and straight-through technique to estimate gradients, which still has some rooms to explore its novelty.

A potential dependency on the diversity of input features, which might limit the framework's generalizability across different markets (e.g., non-US) or datasets with fewer features.

In general, it is an interesting topic to me, and potentially with impacts to the TMLR audiences. Some model interpretability on making the model's predictions and decision-making process would be good to have in-depth analysis.

---

### Decision · Action_Editor_Eu22 · 2024-08-15

**Recommendation:** Accept with minor revision

**Comment:**

The general view of the reviewers is that, although the algorithmic contribution is modest, the work is technically sound and will be of interest to some in the TMLR audience, primarily because stock market prediction is such a challenging real-world application; progress here could have positive repercussions in other critical time series domains.

In their final version, the authors should be sure to incorporate new results and perspectives that were produced during the review process. I would also like to encourage the authors to add a dedicated Limitations section (e.g., directly before the Conclusion) collecting all the limitations mentioned throughout the paper and in the reviewer discussion into a single place. This would be very useful in this kind of paper, since the reader immediately wants to know why they should or shouldn't rush to implement this method in stock trading today.

---AE

**Audience:**

Yes

**Claims And Evidence:**

Yes